# Precise assembly of complex beta sheet topologies from de novo designed building blocks

**Indigo Chris King[1]\*[†], James Gleixner[1], Lindsey Doyle[2], Alexandre Kuzin[3], John F Hunt[3], Rong Xiao[4], Gaetano T Montelione[4], Barry L Stoddard[2], Frank DiMaio[1], David Baker[1]**

[1]Institute for Protein Design, University of Washington, Seattle, United States; [2]Basic Sciences, Fred Hutchinson Cancer Research Center, Seattle, United States; [3]Biological Sciences, Northeast Structural Genomics Consortium, Columbia University, New York, United States; [4]Center for Advanced Biotechnology and Medicine, Department of Molecular Biology and Biochemistry, Northeast Structural Genomics Consortium, Rutgers, The State University of New Jersey, Piscataway, United States

**Abstract** Design of complex alpha-beta protein topologies poses a challenge because of the large number of alternative packing arrangements. A similar challenge presumably limited the emergence of large and complex protein topologies in evolution. Here, we demonstrate that protein topologies with six and seven-stranded beta sheets can be designed by insertion of one de novo designed beta sheet containing protein into another such that the two beta sheets are merged to form a single extended sheet, followed by amino acid sequence optimization at the newly formed strand-strand, strand-helix, and helix-helix interfaces. Crystal structures of two such designs closely match the computational design models. Searches for similar structures in the SCOP protein domain database yield only weak matches with different beta sheet connectivities. A similar beta sheet fusion mechanism may have contributed to the emergence of complex beta sheets during natural protein evolution.

**\*For correspondence:** chrisk1@uw.edu

**Present address:** [†]Molecular Engineering and Sciences, University of Washington, Seattle, United States

**Competing interests:** The authors declare that no competing interests exist.

## Introduction

Modular domains constitute the primary structural and functional units of natural proteins. Multidomain proteins likely evolved through simple linear concatenation of successive domains onto the polypeptide chain or through the insertion of one or more continuous sequences into the middle of another, now discontinuous domain (*Aroul-Selvam et al., 2004*; *Berrondo et al., 2008*; *Lupas et al., 2001*; *Pandya et al., 2013*). By analogy, new proteins have been engineered from existing domains by simple linear concatenation or insertion of one domain into another (*Ay et al., 1998*; *Collinet et al., 2000*; *Cutler et al., 2009*; *Doi and Yanagawa, 1999*; *Edwards et al., 2008*; *Guntas and Ostermeier, 2004*; *Ostermeier, 2005*). How individual domains evolved, in contrast, is much less clear. Both experimental and computational analyses have suggested that new folds can evolve by insertion of one fold into another (*Lupas et al., 2001*; *Grishin, 2001*; *Söding and Lupas, 2003*; *Krishna and Grishin, 2005*; *Friedberg and Godzik, 2005*; *Ben-Tal and Kolodny, 2014*), but to our knowledge, there is no evidence that complex beta sheet topologies can be formed in this manner. On the protein design front, there has been progress in de novo design of idealized helical bundles (*Park et al., 2015*) and alpha beta protein structures with up to 5 strands (*Koga et al., 2012*), and although new folds have been generated by tandem fusion of natural protein domains

**eLife digest** A protein is made up of a sequence of amino acids and must fold into a specific three-dimensional structure if it is to work correctly. The structure is formed by segments of the protein adopting specific shapes, the two most common shapes being alpha helices and beta strands. Beta strands commonly interact with each other to form regions called beta sheets.

Researchers trying to design proteins with new abilities have managed to create proteins that contain up to five beta strands and four alpha helices. Larger and more complex proteins are more challenging to make because there are many different ways that a protein can fold. It is also difficult to understand how complex structures such as large beta sheets emerged naturally, over the course of evolution.

King et al. have now used computer modeling to explore how a large, complex beta sheet might form. In the model, one small, newly designed protein was inserted into another so that their beta sheets merged to form a single extended sheet. The model then stabilized this structure by changing the amino acids found at the points where the two proteins met.

King et al. were then able to synthesize these new proteins in bacteria and use a technique called X-ray crystallography to determine the structure of two of them. The structures closely matched the computer models; one protein contained a six-stranded beta sheet, and the other had a seven-stranded beta sheet. The folds of the two designed proteins were then compared with those found in a database that classifies proteins on the basis of their structure. The beta sheets in the designed proteins did not match the protein structures in the database, which suggests that the designed proteins contained new types of folds.

In the future, the technique used by King et al. could be used to design other large and complex beta sheet structures. Furthermore, the results suggest that such large structures could have evolved naturally through the combination of smaller, less complex proteins.

followed by introduction of additional stabilizing mutations (*Hocker et al., 2004*; *Shanmugaratnam et al., 2012*), assembly of large and complex beta sheets poses a challenge for de novo protein design.

One possible route to the large and complex beta sheet topologies found in many native protein domains is recombination of two smaller beta sheet domains. Here, we explore the viability of such a mechanism by inserting one de novo designed alpha beta protein into another such that the two beta sheets are combined into one. The backbone geometry at the junctions between the original domains is regularized, and the sequence at the newly formed interface is optimized to stabilize the single integrated domain structure. Crystal structures of two such proteins demonstrate that complex beta sheet structures can be designed with considerable accuracy using this approach and provide a proof-of-concept for the hypothesis that complex beta topologies in natural proteins may have evolved from simpler beta sheet structures in a similar manner.

## Results

A first extended sheet protein was created by inserting a designed ferredoxin domain into a beta turn of the designed TOP7 protein to create a half-barrel structure, with the two sheets fused into a single seven strand sheet flanked by four helices (*Figure 1A*). The CD spectra show both alpha and beta structures (*Figure 2—figure supplement 1*). Two crystal structures (NESG target OR327) were solved by molecular replacement and refined to 2.49 Å (PDB entry 4KYZ) and 2.96 Å (PDB entry 4KY3) resolutions. Further analysis refers only to the higher resolution structure (4KYZ). The structure shows excellent agreement with the design model (*Figure 2A*), particularly in low B-factor regions, with C-alpha RMSD ranging from 1.76 to 1.85 Å among the four protomers in the crystal. The relative orientation of the strands packed against the helices is close to that in the design model, and core sidechains at the designed interfaces are in very similar conformations in the design model and crystal (*Figure 2B,C*).

A second extended sheet protein was created by inserting one designed ferredoxin domain into another to create a half-barrel structure with four alpha helices and six beta strands (*Figure 1B*). A

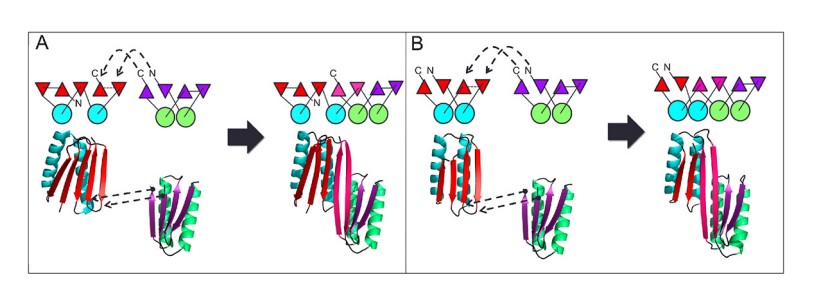

**Figure 1.** Generation of protein domains with single extended beta sheets by inserting one beta sheet containing protein into another. (**A**) Insertion of a ferrredoxin domain (purple) into TOP7 (red). (**B**) Insertion of one ferrodoxin domain into another. In both cases, two beta strands from each partner (red and purple) are concatenated to form the central strand pair of the fusion protein (pink).

beta turn segment between two beta strands of the host ferredoxin was removed and the resulting cut-points in the host beta strands were linked to two beta strand cut-points in the insert, fusing the two strand pairs into a single, longer pair at the center of a six-stranded beta sheet. CD spectra show that the protein contains both alpha and beta structures (*Figure 3—figure supplement 1*). Crystals were obtained which diffracted to 3.3Å resolution. Molecular replacement using the computational design models (*DiMaio et al., 2013*) yielded a solution for which the refinement statistics are shown in *Supplementary file 1* (PDB entry 5CW9). Attempts to improve these statistics by rebuilding portions of the model proved unsuccessful, possibly due to a register shift or dynamic fluctuations in the structure (perhaps corresponding to slightly 'molten-globule'-like behavior) that are difficult to computationally model. However, unbiased low-resolution omit maps suggest that the overall topology is correct (*Figure 3—figure supplement 2*). In the model that displays the best refinement statistics, the protein backbone was similar to the design model with a C-alpha RMSD value of 2 Å (*Figure 3A,B*). The fused beta sheet aligns with the design model, while the inter-domain helices shift slightly to accommodate the inter-domain interface. The sidechain packing between the newly juxtaposed beta strands succeeded in anchoring the secondary structure

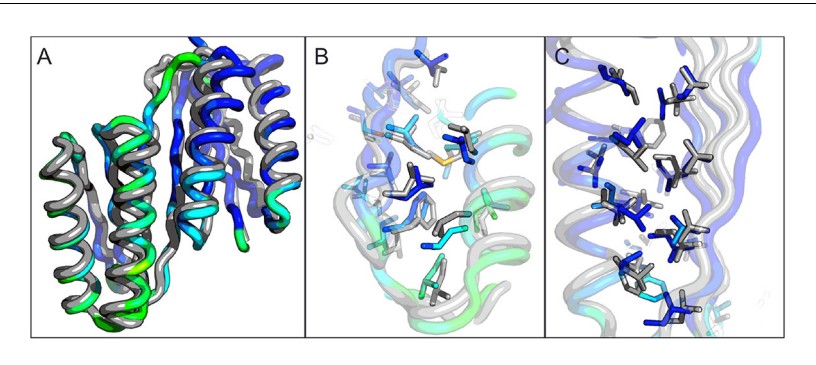

**Figure 2.** Comparison of the crystal structure of ferrodoxin-TOP7 fusion to design model. (**A**) Backbone superposition of the crystal structure of ferrodoxin-TOP7 (4KYZ, chain A) with the design model. The backbones of the two proteins are nearly identical. (**B, C**) The core sidechain packing in the ferrodoxin-TOP7 fusion is very similar in the crystal structure and design model both in the insert (**B**) and host (**C**) domains. The crystal structure is colored by B-factor and the design model is in gray.

The following figure supplement is available for figure 2:

**Figure supplement 1.** The circular dichroism spectrum of ferrodoxin-TOP7 has the shape expected for an alpha/beta protein.

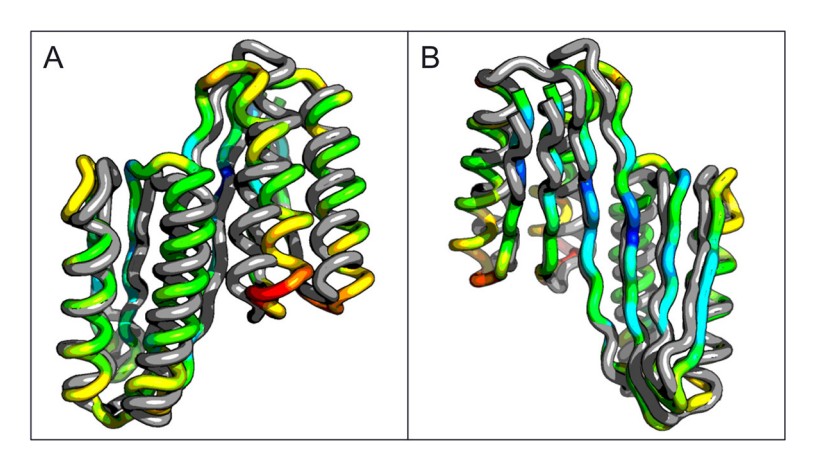

**Figure 3.** Comparison of the crystal structure of the ferredoxin-ferredoxin fusion to the design model. The crystal structure (5CW9) aligns well with the design model over both the helices (**A**) and the fused beta sheet (**B**).

The following figure supplements are available for figure 3:

**Figure supplement 1.** Circular dichroism spectra of ferrrodoxin-ferrodoxin at 25°C.

**Figure supplement 2.** Ferredoxin-Ferredoxin 2Fo-Fc omit map superimposed with crystal structure shows core packing of host (**A**) and insert (**B**) domains.

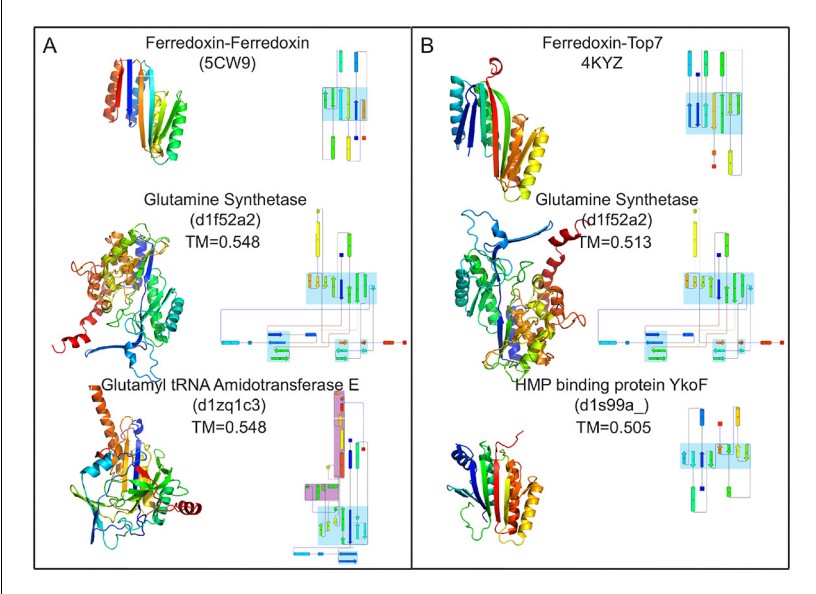

**Figure 4.** Top two SCOP domain structural homologues for Fd-Top7 (**A**) and Fd-Fd (**B**) designed domain found in TM-align searches. Ribbon diagrams are shown on left, the strand connectivity, at the right. The beta strand connectivity is quite different in the designs than in these closest structural matches.

The following figure supplements are available for figure 4:

**Figure supplement 1.** Parent domain PDB structures (2KL8, 1QYS) and daughter designed folds (5CW9,4KYZ) (pink) mapped into the $\alpha+\beta$ region of the SCOP domains network of Nepomnyachi et al. (**A**) and zoomed region (**B**) highlighting parent, designed, and first neighbor folds.

**Figure supplement 2.** Neutral drift mutant models, relative changes to predicted free energy of folding in REU (Rosetta Energy Units), and multiple sequence alignment of parent and designed sequences, showing mutations in ferredoxin-top7 (**A**) and ferredoxin-ferredoxin (**B**).

elements in their intended orientations, but the low resolution of the crystal structure prevents evaluation of the atomic-level accuracy of the design (*Figure 3—figure supplement 2*).

To compare the folds of these designed proteins to those in the SCOP v.1.75 domain database (*Murzin et al., 1995*), the TMalign structure-structure comparison method was used to search a 70% sequence non-redundant set of SCOP domains (*Ben-Tal and Kolodny, 2014*) for structure alignments containing a minimum 75% overlap with the designed proteins. The most similar SCOP domains had weak TM-align scores (0.54 and 0.51), and the sheets in these matched structures have different connectivities than those of the designs, suggesting that the two designed proteins have novel folds (*Figure 4*). While there are no domains with globally similar folds, both designed proteins are similar to a number of SCOP domains over the ferrodoxin-like substructure(s) (maps of the proteins to the domain network of Nepomnyachiy et al. (*Ben-Tal and Kolodny, 2014*) are shown in *Figure 4—figure supplement 1*). The mutations introduced at the redesign stage of the domain insertion design protocol are compatible with the parent fold structures with minimal perturbation of the protein backbone (*Figure 4—figure supplement 2*) suggesting the designed folds would have the potential to evolve from insertion followed by neutral mutational drift of the parent structures.

## Discussion

We have shown that single designed protein domains can be combined into larger domains with complex beta sheet topologies. This mechanism provides a straightforward route to designing large and complex beta sheet structures capable of scaffolding the pockets and cavities essential for future design of protein functions. Our success in designing larger beta sheet domains by recombining smaller independently folded beta sheet proteins suggests a similar mechanism could have played a role in the evolution of naturally occurring complex beta sheet proteins.

## Materials and methods

Our design strategy began with selection of three previously characterized de novo designed protein domains to serve as building blocks for recombination through domain insertion: ferredoxin, rossman 2x2, and top7 (*Koga et al., 2012*). These three domains were chosen because they were the only Rosetta de novo designed protein domains with both alpha and beta secondary structures for which high-resolution experimental structures had been obtained at the time of this work. Each chimeric domain consists of a parent host domain and a parent insert domain. In the insert domain, three residues from from the N-terminus were paired with three residue from the C-terminus to create nine residue pairs. Each residue pair was then aligned against all pairs of residues in the host domain to search for possible insertion points. Insertion points were accepted for residue pair alignment distances of 1 angstrom RMSD or less, replacing host domain segments of less than 5 residues. For every insertion point, a structure is generated by removing the residues between the insertion residues of the host domain and adding linkers between the aligned host and insert domain residues (*Figure 1*). Host and insert were connected by addition of 1–3 residues at the domain junctions using Rosetta Remodel (*Huang et al., 2011*), and 12 models in which this junction formed a continuous beta strand were identified. The sequences of these chimeras were optimized using Rosetta Design calculations around the junction regions and the new interface between the former domains. During the design simulation, all amino acid positions within 5 Å of the inter-domain junction interface were redesigned to minimize the predicted free energy of folding with the Rosetta all-atom energy function and a flexible backbone protein design protocol described previously (*Huang et al., 2011*). Final designs were selected based on Rosetta energy, packing metrics, and similarity of the junction backbone geometry to local backbone geometry in the PDB. Twelve final domain insertion designs were chosen for expression in *Escherichia coli* as 6xHis-tag fusions and purified on a Ni-NTA column. Purified proteins were evaluated for the presence of alpha/beta secondary structures via circular dichroism spectroscopy (CD), and three with levels of secondary structure content consistent with the design model were subjected to crystallographic analysis. One design based on Rossman 2x2 expressed as soluble protein, but no crystal structure could be obtained. Crystal structures were obtained for two designed proteins: a ferredoxin-top7 chimera and a ferredoxin-ferredoxin chimera. The design and characterization of these two proteins is described in the Results.

Crystal structures were used to search for structural homologs in the SCOP database. First, crystal structures (ferredoxin-top7: 4KYZ chain A, ferredoxin-ferredoxin: 5CW9 chain A) were used as search queries using TMalign (*Zhang, 2005*). Hits were saved only if the alignment covered 75% or more of the query structure. Results were sorted by TM-score to identify the most similar structures in the SCOP database. Secondary structure topology cartoons were created with the Pro Origami server (*Stivala et al., 2011*). To map designed protein crystal structures into the protein domains network, the structures were aligned to all domain structures in the protein domains network using the PDBe-Fold server (*Krissinel and Henrick, 2004*). PDBeFold structural alignment hits were filtered for RMSD ≤2.5 Å and aligned sequence length of ≥75 residues. In contrast to the methods of Nepomnyachi et al., sequence similarity thresholds were ignored. Including sequence similarity thresholds eliminates matching hits in the domains network. This is not surprising because the proteins were designed de novo and did not evolve from natural proteins. Filtered alignment hits were mapped into the protein domains network using Cytoscape (*Shannon, 2003*). To evaluate neutral drift models of the parent folds, then crystal structures of de novo ferredoxin and Top7 proteins (2KL8 and 1QYS) were obtained and corresponding mutations from the final design proteins were modeled using a flexible backbone protein design algorithm described previously (*Huang et al., 2011*). Final Rosetta energies were calculated and subtracted from the Rosetta energies of the original parent protein structures to obtain predictions of the change in free energy of folding.

The ferredoxin – TOP7 protein (NESF ID OR327) was expressed and purified following standard protocols developed by the NESG for production of selenomethionine-labeled protein samples (*Xiao et al., 2010*). Briefly, *E. coli* BL21 (DE3) pMGK cells, a rare-codon enhanced strain, were transformed with the DNA sequence-verified OR327-21.1 plasmid. A single isolate was cultured in MJ9 minimal media supplemented with selenomethionine, lysine, phenylalanine, threonine, isoleucine, leucine, and valine for the production of selenomethionine-labeled OR327. Initial growth was carried out at 37°C until the OD600 of the culture reached ~0.8 units. The incubation temperature was then decreased to 17°C, and protein expression was induced by the addition of isopropyl-β-D-thiogalactopyranoside (IPTG) at a final concentration of 1 mM. Following overnight incubation at 17°C, the cells were harvested by centrifugation and resuspended in Lysis Buffer [50 mM Tris, pH 7.5, 500 mM NaCl, 1 mM tris (2-carboxyethyl)phosphine, 40 mM imidazole]. After sonication, the supernatant was collected by centrifugation for 40 min at 30,000 *g*. The supernatant was loaded first onto a Ni affinity column (HisTrap HP; GE Healthcare, Marlborough, MA) and the eluate loaded into a gel filtration column (Superdex 75 26/60; GE Healthcare). Yields were 60-–90 mg/L. The purified 6His-OR327 construct in buffer containing 10 mM Tris·HCl, 100 mM NaCl, 5 mM DTT, pH 7.5, was then concentrated to ~10.6 mg/mL. The sample was flash-frozen in 50-μL aliquots using liquid nitrogen and stored at −80°C before crystallization trials. The sample purity (>98%), molecular weight, and oligomerization state were verified by SDS/PAGE, MALDI-TOF mass spectrometry, and analytic gel filtration followed by static light scattering, respectively. For static light scattering, selenomethionine-labeled ferredoxin – TOP7 protein (30 μL at 10 mM Tris·HCl, pH 7.5, 100 mM NaCl, 5 mM DTT) was injected onto an analytical gel filtration column (Shodex KW-802.5; Shodex, New York, NY) with the effluent monitored by refractive index (Optilab rEX; Wyatt Technology, Santa Barbara, CA) and 90° static light-scattering (miniDAWN TREOS; Wyatt Technology) detectors.

## Accession codes

Structures have been deposited in the Protein Data Bank as entries 5CW9, 4KYZ, and 4KY3.

## Acknowledgements

We thank Rie Koga and Nobuyasu Koga for data analyses and technical assistance. We thank Lei Mao for technical assistance. This work was supported by the Defense Threat Reduction Agency and by a grant from the National Institute of General Medical Sciences Protein Structure Initiative U54-GM094597 (to GTM, JH).

## Additional information

### Funding

| Funder | Author |
| --- | --- |
| Defense Threat Reduction Agency | Indigo Chris King James Gleixner David Baker |
| National Institute of General Medical Sciences | John F Hunt Gaetano T Montelione |

The funders had no role in study design, data collection and interpretation, or the decision to submit the work for publication.

### Author contributions

ICK, wrote code for simulations, performed simulations, and wrote the paper; commented on the manuscript, Conception and design, Acquisition of data, Analysis and interpretation of data, Drafting or revising the article; JG, performed simulations, performed experiments, and analyzed data; commented on the manuscript, Conception and design, Acquisition of data, Analysis and interpretation of data, Drafting or revising the article; LD, BLS, participated in crystallization of the Fd-Fd construct and subsequent data collection and data processing; commented on the manuscript, Acquisition of data, Analysis and interpretation of data, Drafting or revising the article; AK, JFH, solved the Fd-Top7 crystal structure; commented on the manuscript, Acquisition of data, Analysis and interpretation of data, Drafting or revising the article; RX, GTM, expressed and purified protein samples; commented on the manuscript, Acquisition of data, Analysis and interpretation of data, Drafting or revising the article; FDiM, conducted molecular replacement analyses and subsequent refinement for the Fd-Fd crystal structure using an ensemble of molecular search models produced by RosettaDesign; commented on the manuscript, Acquisition of data, Analysis and interpretation of data, Drafting or revising the article; DB, wrote code for simulations, performed simulations, and wrote the paper; commented on the manuscript, Conception and design, Analysis and interpretation of data, Drafting or revising the article

## Additional files

### Supplementary files

• Supplementary file 1. Crystallographic data.

### Major datasets

The following datasets were generated:

| Author(s) | Year | Dataset title | Dataset URL | Database, license, and accessibility information |
| --- | --- | --- | --- | --- |
| DiMaio F, King IC, Gleixner J, Doyle L, Stoddard B, Baker D | 2015 | Crystal structure of De novo designed ferredoxin-ferredoxin domain insertion protein | http://www.rcsb.org/pdb/explore/explore.do?structureId=5CW9 | Publicly available at the RCSB Protein Data Bank (Accession no: 5CW9). |
| Kuzin A, Su M, Seetharaman J, Maglaqui M, Xiao R, Lee D, Gleixner J, Baker D, Everett JK, Acton TB, Kornhaber G, Montelione GT, Hunt JF, Tong L, Northeast Structural Genomics Consortium | 2013 | Three-dimensional Structure of the orthorhombic crystal of computationally designed insertion domain , Northeast Structural Genomics Consortium (NESG) Target OR327 | http://www.rcsb.org/pdb/explore/explore.do?structureId=4KYZ | Publicly available at the RCSB Protein Data Bank (Accession no: 4KYZ). |

| Kuzin A, Su M, Seetharaman J, Maglaqui M, Xiao R, Lee D, Gleixner J, Baker D, Everett JK, Acton TB, Montelione GT, Tong L, Hunt JF, Northeast Structural Genomics Consortium | 2013 | Three-dimensional Structure of the orthorhombic crystal of computationally designed insertion domain , Northeast Structural Genomics Consortium (NESG) Target OR327 | http://www.rcsb.org/pdb/explore/explore.do?structureId=4KY3 | Publicly available at the RCSB Protein Data Bank (Accession no: 4KY3). |
|---|---|---|---|---|

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
