## [Decision Letter]

Thank you for submitting your work entitled "Precise Assembly of Complex Beta Sheet Topologies from de novo Designed Building Blocks" for peer review at *eLife*. Your submission has been evaluated by John Kuriyan (Senior editor) and three reviewers, one of whom, Nir Ben-Tal, served as guest Reviewing editor.

The reviewers have discussed the reviews with one another and the Reviewing editor has drafted this decision that summarizes the discussion.

Summary:

The manuscript describes the design of large alpha/beta domains by fusion of two smaller domains. Crystal structures of two of the designed proteins showed agreement with the design. The successful design supports the suggestion that proteins may evolve by cut-and-paste, where complex domains emerge by the assembly of shorter fragments.

Based on the current draft of the manuscript it is difficult to decide whether the work is novel enough to justify publication in *eLife*. This letter reflects a long discussion among the reviewers in an effort to realize possible novelties. The authors should submit a revised draft only if they are certain that they can address all the issues that were raised.

Essential revisions:

Novelty here could be related to the successful design, as well as the implications to protein evolution. Regarding design, the authors claim in the Introduction that "while protein domains with larger and more complex beta sheets occur frequently in nature, such topologies have not been successfully created by de novo protein design". However, this statement is wrong. For example, Birte Höcker has made a nine-stranded alpha-beta barrel by fusing half of a TIM barrel with most of a response regulator domain, then used design to revert this barrel to the dominant eight-stranded form with five mutations, which could have been sampled by neutral drift (PNAS 2008, JACS 2012). This statement should be corrected accordingly (maybe there are more examples?).

However, there are three other possible novelties here:

1) The fusion of two full, independent domains into a single new domain. Previous efforts have fused fragments of domains. Here it is essential to know more about the new domains. Are the new domains folding cooperatively? Where are they in the list of twelve top ranked constructs (and how was that ranking done)? Near the top, or is there little correlation between ranking and success? What are the biophysical properties of the ten constructs that did not yield structures? A multiple alignment of the twelve experimentally tried constructs compared to the parent proteins would be really useful.

2) Fusion as an insertion of one domain into another. To the best of our knowledge, other cases are all consecutive, N-to-C fusions (but please check).

3) Generation of a fold that is different from that of the parent proteins. To the best of our knowledge, previous examples, like Riechmann and Winter's cold-shock protein fusions, Tawfik's tachylectin propellers, Blaber's beta-trefoils, ended up at or near the parent fold (in fact, most were intended to). The folds in this paper are clearly different from the starting folds. How different are they from other folds in the database? Do they occur in any of the 60 odd SCOP superfamilies that have a ferredoxin-like core (often with elaborate decorations)? The authors should discuss the folds they have generated. Perhaps map the parent and designed folds into the domains network of Nepomnyachiy et al. PNAS 111, 1691-11696, 2014.

Of these 3 novelties, only the third justifies publication in *eLife*, provided that the new fold differs significantly from the original ones.

4) Another source of novelty may come from the implications of the successful design to evolution. The manuscript tries to sell this view, but much more effort is required to substantiate it. The question that emerges is whether evolution can follow the path taken by the design approach. Two issues are outstanding here: first, only 2 of 12 designs have been successful. Second, interface redesign was required and it is unclear whether the normal evolutionary drift would suffice for that. So the question is whether evolution uses the possibility of fusion and the authors need to address this issue. Specifically, they should:

A) Identify examples where a new fold has arisen from the seamless merger of two simpler folds.

B) Add a lucid explanation of the mutations introduced at the redesign stage (with a multiple alignment figure). Nature does not do redesign, so the question is how many changes are needed to make the daughter protein viable and are these changes neutral for the viability of the parent proteins (i.e. could these mutations have been sampled by neutral drift)?

C) Cite related publications by Godzik, Grishin, Soding, Lupas, Ben-Tal, Kolodny, and others.

In addition, the revised draft should address the following issues:

5) The crystallographic information concerning the accuracy of the solutions seems inadequate to judge whether the structures were in fact correctly determined. The revised manuscript should provide additional information concerning the structure determinations and refinements to let the informed reader judge that the solutions are robust.

6) Discussion: "This mechanism provides a straightforward route to designing large and complex beta sheet structures capable of hosting functions such as catalysis difficult to achieve in previously designed small protein domains which lack cavities for potential active sites, etc." The connection between the data shown here and design for function is not obvious. The authors need to elaborate on this. Anyway, this speculation should be tuned down significantly.

7) The manuscript clarifies the design approach, but details about the methodology used are missing. For example, what were the considerations for selecting the ferredoxin and top7 as starting points? There should be a section that explains the computational methodology. Among other things this section should address the following questions and concerns:

A) The description says that also Rossman 2x2 was used in addition to ferredoxin and top7. What happened to designs based on Rossman 2x2?

B) "Positions within two residues of each of the termini of each domain were aligned with all pairs of residues separated by fewer than 5 residues in each domain". The authors should elaborate on this. Maybe add a figure?

C) "The sequences of these chimeras were optimized using Rosetta Design calculations around the junction regions". Optimized for what?

---

## [Author Response]

Essential revisions:

*Novelty here could be related to the successful design, as well as the implications to protein evolution. Regarding design, the authors claim in the Introduction that "while protein domains with larger and more complex beta sheets occur frequently in nature, such topologies have not been successfully created by de novo protein design". However, this statement is wrong. For example, Birte Höcker has made a nine-stranded alpha-beta barrel by fusing half of a TIM barrel with most of a response regulator domain, then used design to revert this barrel to the dominant eight-stranded form with five mutations, which could have been sampled by neutral drift (PNAS 2008, JACS 2012). This statement should be corrected accordingly (maybe there are more examples?).* The claim has been modified in the text with related citations of Höcker et al:

“On the protein design front, there has been progress in de novo design of idealized helical bundles (Park et al., 2015) and alpha beta protein structures with up to 5 strands (Koga et al., 2012), and though new folds have been generated by tandem fusion of natural protein domains followed by introduction of additional stabilizing mutations (Hocker, Claren and Sterner, 2004; Shanmugaratnam, Eisenbeis and Hocker, 2012), assembly of large and complex beta sheets poses a challenge for de novo protein design.”

*However, there are three other possible novelties here:*

*1) The fusion of two full, independent domains into a single new domain. Previous efforts have fused fragments of domains. Here it is essential to know more about the new domains. Are the new domains folding cooperatively? Where are they in the list of twelve top ranked constructs (and how was that ranking done)? Near the top, or is there little correlation between ranking and success? What are the biophysical properties of the ten constructs that did not yield structures? A multiple alignment of the twelve experimentally tried constructs compared to the parent proteins would be really useful.*

*2) Fusion as an insertion of one domain into another. To the best of our knowledge, other cases are all consecutive, N-to-C fusions (but please check). 3) Generation of a fold that is different from that of the parent proteins. To the best of our knowledge, previous examples, like Riechmann and Winter's cold-shock protein fusions, Tawfik's tachylectin propellers, Blaber's beta-trefoils, ended up at or near the parent fold (in fact, most were intended to). The folds in this paper are clearly different from the starting folds. How different are they from other folds in the database? Do they occur in any of the 60 odd SCOP superfamilies that have a ferredoxin-like core (often with elaborate decorations)? The authors should discuss the folds they have generated. Perhaps map the parent and designed folds into the domains network of Nepomnyachiy et al. PNAS 111, 1691-11696, 2014. Of these 3 novelties, only the third justifies publication in* eLife*, provided that the new fold differs significantly from the original ones.*

As the reviewers have suggested, we have focused on novelty (3) above in the manuscript and elaborated on the uniqueness of the designed fold in the new third section of the Results text and in two additional figures (Figure 4 and Figure 4—figure supplement 1). We have included an analysis of the similarity of the designed protein to those in SCOP in Figure 4 (TM-align hits across the entire new fold) and mapped the designed protein domains and their parent folds into the protein domains network in Figure 4—figure supplement 1:

“To compare the folds of these designed proteins to those in the SCOP v.1.75 domain database (Murzin et al., 1995), the TMalign structure-structure comparison method was used to search a 70% sequence non-redundant set of SCOP domains for structure alignments containing a minimum 75% overlap with the designed proteins. […] While there are no domains with globally similar folds, both designed proteins are similar to a number of SCOP domains over the ferrodoxin-like substructure(s) as is made evident by mapping the proteins to the domains network of Nepomnyachiy et al. (Ben-Tal and Kolodny, 2014) (Figure 4—figure supplement 1).”

*4) Another source of novelty may come from the implications of the successful design to evolution. The manuscript tries to sell this view, but much more effort is required to substantiate it. The question that emerges is whether evolution can follow the path taken by the design approach. Two issues are outstanding here: first, only 2 of 12 designs have been successful. Second, interface redesign was required and it is unclear whether the normal evolutionary drift would suffice for that. So the question is whether evolution uses the possibility of fusion and the authors need to address this issue. Specifically, they should:*

*A) Identify examples where a new fold has arisen from the seamless merger of two simpler folds.*

Previous studies identifying the evolution of new folds from the combination of multiple simple folds have been cited in the Introduction:

“By analogy, new proteins have been engineered from existing domains by simple linear concatenation or insertion of one domain into another (Ay et al., 1998; Collinet et al., 2000; Cutler et al., 2009; Doi and Yanagawa, 1999; Edwards et al., 2008; Guntas and Ostermeier, 2004; Ostermeier, 2005). How individual domains evolved, in contrast, is much less clear. Both experimental and computational analyses have suggested that new folds can evolve by insertion of one fold into another (Lupas, Ponting and Russell, 2001; Grishin, 2001; Soding and Lupas, 2003; Krishna and Griffin, 2005; Friedberg and Godzik, 2005; Bel-Tal and Kolodny, 2014), but to our knowledge there is no evidence that complex beta sheet topologies can be formed in this manner.”

*B) Add a lucid explanation of the mutations introduced at the redesign stage (with a multiple alignment figure). Nature does not do redesign, so the question is how many changes are needed to make the daughter protein viable and are these changes neutral for the viability of the parent proteins (i.e. could these mutations have been sampled by neutral drift)?*

We have added a third section to the Results and an additional figure (Figure 4—figure supplement 2) showing a multiple sequence alignment, computational models of the putative neutral drift parent fold mutants, and calculations predicting the energetic effects of these neutral drift mutations on the parent folds:

“The mutations introduced at the redesign stage of the domain insertion design protocol are compatible with the parent fold structures with minimal perturbation of the protein backbone (Figure 4—figure supplement 2) suggesting the designed folds would have the potential to evolve from insertion followed by neutral mutational drift of the parent structures.”

We have elaborated the explanation of the mutations introduced at the design stage in the Methods, and the location of the redesign mutations are shown in Figure 4—figure supplement 2:

“During the design simulation, all amino acid positions within 5 Å of the inter-domain junction interface were redesigned to minimize the predicted free energy of folding with a flexible backbone protein design protocol described previously (Friedberg and Godzik, 2005).”

*C) Cite related publications by Godzik, Grishin, Soding, Lupas, Ben-Tal, Kolodny, and others.*

The above publications have been cited in the Introduction.

*In addition, the revised draft should address the following issues:*

*5) The crystallographic information concerning the accuracy of the solutions seems inadequate to judge whether the structures were in fact correctly determined. The revised manuscript should provide additional information concerning the structure determinations and refinements to let the informed reader judge that the solutions are robust.*

Key structure quality assessment statistics for the crystal structures are presented in the table of [Supplementary-material SD1-data]. In addition, we attach in these responses to reviewers wwPDB Structure Validation Reports. These reports, generated by the wwPDB, follow recommendations of the wwPDB X-ray Crystallography Structure Validation Task Force. The evolving standard of the field is to provide these reports to reviewers to address exactly the concerned raised by Reviewer 1.

We have added additional clarifications in the Results section to address the crystal structure accuracy, and have made it more clear that in the ferredoxin-ferredoxin structure, the fine details of the structure difficult to model, but the overall topology is not in doubt. Conversely, in the ferredoxin-top7 crystal (ferredoxin-top7), structure refinement was well-behaved.

“Attempts to improve these statistics by rebuilding portions of the model proved unsuccessful, possibly due to a register shift or dynamic fluctuations in the structure (perhaps corresponding to slightly 'molten-globule'-like behavior) that are difficult to computationally model. However, unbiased low-resolution omit maps suggest that the overall topology is correct (Figure 2—figure supplement 2).”

*6) Discussion: "This mechanism provides a straightforward route to designing large and complex beta sheet structures capable of hosting functions such as catalysis difficult to achieve in previously designed small protein domains which lack cavities for potential active sites, etc." The connection between the data shown here and design for function is not obvious. The authors need to elaborate on this. Anyway, this speculation should be tuned down significantly.*

The speculation has been changed to:

“This mechanism provides a straightforward route to designing large and complex beta sheet structures capable of scaffolding the pockets and cavities essential for future design of protein functions.”

*7) The manuscript clarifies the design approach, but details about the methodology used are missing. For example, what were the considerations for selecting the ferredoxin and top7 as starting points? There should be a section that explains the computational methodology. Among other things this section should address the following questions and concerns:*

The following sentence has been added to the Methods:

“These three domains were chosen because they were the only Rosetta de novo designed protein domains with both alpha and beta secondary structure for which high resolution experimental structures had been obtained.”

*A) The description says that also Rossman 2x2 was used in addition to ferredoxin and top7. What happened to designs based on Rossman 2x2?*

The following sentence has been added to the Methods:

“One design based on Rossman 2x2 expressed as soluble protein, but no crystal structure could be obtained at the time of this work.”

*B) "Positions within two residues of each of the termini of each domain were aligned with all pairs of residues separated by fewer than 5 residues in each domain". The authors should elaborate on this. Maybe add a figure?*

The insertion point and design protocols have been clarified and elaborated in the Methods:

“Each chimeric domain consists of a parent host domain and a parent insert domain. […] During the design simulation, all amino acid positions within 5 Å of the inter-domain junction interface were redesigned to minimize the predicted free energy of folding with the Rosetta all-atom energy function and a flexible backbone protein design protocol described previously (Friedberg and Godzik, 2005).”

*C) "The sequences of these chimeras were optimized using Rosetta Design calculations around the junction regions". Optimized for what?*

The following sentence has been added to the Methods:

“The sequences of these chimeras were optimized using Rosetta Design calculations around the junction regions and the new interface between the former domains. During the design simulation, all amino acid positions within 5 Å of the inter-domain junction interface were redesigned to minimize the predicted free energy of folding with the Rosetta all-atom energy function and a flexible backbone protein design protocol described previously (Friedberg and Godzik, 2005).”